# Implications of Extended Inhibitory Neuron Development

**DOI:** 10.3390/ijms22105113

**Published:** 2021-05-12

**Authors:** Jae-Yeon Kim, Mercedes F. Paredes

**Affiliations:** 1Department of Neurology, University of California, San Francisco, CA 94143, USA; 2Eli and Edythe Broad Center of Regeneration Medicine and Stem Cell Research, University of California, San Francisco, CA 94143, USA; 3Neuroscience Graduate Division, University of California, San Francisco, CA 94143, USA

**Keywords:** GABAergic inhibitory neuron, embryonic neurogenesis, postnatal migration, functional network, gyrencephalic brain

## Abstract

A prolonged developmental timeline for GABA (γ-aminobutyric acid)-expressing inhibitory neurons (GABAergic interneurons) is an amplified trait in larger, gyrencephalic animals. In several species, the generation, migration, and maturation of interneurons take place over several months, in some cases persisting after birth. The late integration of GABAergic interneurons occurs in a region-specific pattern, especially during the early postnatal period. These changes can contribute to the formation of functional connectivity and plasticity, especially in the cortical regions responsible for higher cognitive tasks. In this review, we discuss GABAergic interneuron development in the late gestational and postnatal forebrain. We propose the protracted development of interneurons at each stage (neurogenesis, neuronal migration, and network integration), as a mechanism for increased complexity and cognitive flexibility in larger, gyrencephalic brains. This developmental feature of interneurons also provides an avenue for environmental influences to shape neural circuit formation.

## 1. Introduction

GABAergic inhibitory neurons (interneurons) represent approximately 20% of the neuronal population in the mammalian brain but have great diversity in origin, composition, and function [1,2]. Interneurons tightly control signal processing through the secretion of the neurotransmitters, GABA (γ-aminobutyric acid), which are essential for the maintenance of brain homeostasis or delicate processing of information. For example, interneurons locally coordinate the timing and tone of excitatory neuronal firing to generate oscillatory rhythm [3]; the combinational activation of diverse interneuron types creates a selectivity for different sensory inputs, facilitating the discrimination of sensory information [4]. Recent postmortem studies have associated genetic mutations in cortical interneurons, abnormal interneuron density, and decreased levels of GABA with the pathological conditions observed in autism spectrum disorder (ASD), epilepsy, and psychiatric disorders [5,6,7]. Therefore, it is important to understand the normal development of GABAergic interneurons to determine their vulnerability in disease and better target therapeutic interventions.

GABAergic interneurons are made in the ventral embryonic brain, in a region called the ganglionic eminences (GE). This developmental/neurogenic niche has several shared molecular and anatomical features between mice and humans [8,9,10]. For instance, the subcortical GE origins, the transcription factors (TF) expressed in GEs, and the tangential migratory routes of cortical GABAergic interneurons are conserved well from mice to humans [9]. While the extended timeline for the production and development of interneurons is observed across many species [11], the protracted nature of these stages is amplified in larger, gyrencephalic brains [12]. Extending these processes could underlie the expansion and unique composition of interneurons observed in the brains of species such as macaque monkeys and humans [13,14,15]. Furthermore, it could maintain the brain in a more dynamic state for an extended period of time during postnatal stages, permitting enhanced interactions with and sensitivity to new, external stimuli. In this review, we review the postnatal incorporation of GABAergic interneurons into the forebrain of gyrencephalic animals via “sustained” neurogenesis, migration, and potential latent sources of GABAergic interneurons. With this lens, we discuss the molecular mechanisms underlying these expanded stages and how disruption could be reflected in pathological conditions.

## 2. Extended Production of Cortical GABAergic Interneuron

### 2.1. Neurogenesis of Cortical GABAergic Interneuron Extends Until the End of Gestation

Neurogenesis, the production of new neurons, in the embryonic brain is the first step underlying the structural development of the nervous system [16]. Neurogenesis from specific progenitors, or neural stem cells, contributes to the total number of neurons in the brain and may determine brain size [17]. For example, brains from gyrencephalic species, such as ferrets and humans, have an enrichment of unique progenitor cells, termed outer radial glial cells [18]. These cells are highly proliferative when compared to apical or ventricular radial glia cells and are present until mid-gestation at 22 weeks [16]. The abundance of these progenitor cells in the developing human brain is thought to contribute to an increased number and diversity of excitatory neurons. The regional differences in their neurogenic capacity are also suggested to promote the specific folding of the cortical plate to form the folds seen in gyrencephalic brains [19]. Less is known regarding the embryonic neurogenesis of GABAergic interneurons in larger brains and whether interneuron progenitors have similar heterogeneous properties.

Unlike cortical excitatory neurons that are produced in the dorsal embryonic brain, cortical GABAergic interneurons are generated from the ventral subregion called the GE; this area can be further subdivided into the medial ganglionic eminence (MGE), the caudal ganglionic eminence (GCE), and the lateral ganglionic eminence (LGE), which are present during gestation [20,21,22] (Figure 1). The GE subregions can be defined by the expression of spatially restricted TFs, and each area generates unique and distinct interneuron subtypes [23,24]. For instance, the MGE preferentially expresses NKX2.1 and LHX6 and produces specific interneurons, such as somatostatin+ (SST)+ and parvalbumin+ (PV)+ interneurons. In contrast, SP8, COUP-TF2, PROX1, and PAX6 are predominantly expressed in the CGE, which produces Reelin+, vasoactive intestinal peptide+ (VIP)+, and cholecystokinin+ (CCK)+ interneuron subtypes [25,26,27]. In humans, the CGE also contributes to a secretagogin (SCGN)+ population [28]. Less is known about TFs that are specific for the LGE that generates OB neurons, such as tyrosine hydroxylase+ (TH)+ and calretinin+ (CR)+ interneurons [8,26,29]. While the progenitor domains for interneurons are highly conserved from rodents to gyrencephalic species [9], the period of inhibitory neurogenesis in the GEs varies considerably across species [30,31]. In mice, neurogenesis in the MGE takes place from embryonic day 10.5 (E10.5) to E16 [32,33,34,35]. In the human GE, the number of interneural progenitor cells that are proliferative (Ki67+) is reported to peak in the second trimester but are still detected up to 35 GW at the end of gestation [8,36,37]. Strikingly, the rabbit GE has a similar pattern for proliferating cell numbers as human GEs, but the rabbit MGE and CGE retain their proliferative ability even after birth [37,38]. The CGE remains proliferative later than the MGE in both mice and humans, but CGE-derived interneurons make up a larger proportion of the interneuron population in human brains—up to 50% depending on the cortical region [8,9]. Thus, differential neurogenesis of each GE may underlie the generation of the diverse cortical interneuron types and their distinct proportions in the cortex [39]. The protracted period of inhibitory neurogenesis and the distinct expansion of proliferation, such as in the CGE in humans, may lead to the later incorporation of specific GABAergic interneuron subtypes in the postnatal forebrain and the acquisition of cognitive functions mediated by interneurons.

### 2.2. Epigenetic Regulation during Neurogenesis

During embryonic development, neuronal progenitors proliferate under a changing intrauterine environment. For instance, the placenta hormones, including estrogens, progesterone, and human placental lactogens, are some of the major intrauterine stimuli that can influence neurogenesis [40]. One mechanism through which the embryonic environment can regulate the developing nervous system is that of epigenetic modification, which mediates genetic expressions without alteration of the DNA sequence [41]. For example, ambient temperature can change the sex of reptiles, such as turtles, through the regulation of the histone demethylase KDM6B in temperature-sensitive embryonic gonads [42,43]. In mammals, the DNA methylation status of the maternal oxytocin gene promoter during the mid-late gestation of human pregnancy is associated with specific maternal behaviors, such as intrusiveness [44]. Recent epigenetic neurodevelopmental studies in the mouse brain identified that chromatin methylation in neuronal progenitor cells can regulate neurogenesis [45,46]. DNA methylation is catalyzed by an enzymatic family of DNA methyltransferases (DNMTs; DNMT1 and DNMT3). DNMT1 is highly expressed for the maintenance of methylation, both in progenitor and postmitotic neuronal cells. On the other hand, DNMT3 methylates newly synthesized DNA to copy the parent DNA methylation pattern during early development [47,48]. In particular, DNMT3A is enriched by progenitors within the ventricular zones during E10.5–E17.5; it directly methylates cis-regulatory regions for inhibitory neurogenic genes, such as DLX2, SP8, and NEUROG2, thereby promoting neurogenesis to increase the number of interneurons [49,50]. Given that DNMT1 and DNMT3A are highly expressed, especially in cortical GABAergic interneurons in the piriform cortex, motor cortex, hippocampus, and amygdala, it is implicated that DNA modification by DNMT1 and 3A is involved in cortical GABAergic interneurons [51,52]. In contrast, LSD1 (lysine-specific histone demethylase 1A, also known as KDM1) participates in the removal of methylation in histone, a component of chromatin. It demethylates at Lysine 9 on Histone H3 at cell cycle arrest genes (e.g., *p21* and *Pten*) to downregulate their levels, thereby maintaining neurogenesis [53]. Interestingly, LSD1 specifically regulates neuronal differentiation through the modulation of the promoter regions of *NOTCH* genes in human fetal progenitors, a region not observed in mouse progenitors [54]. Additionally, human fetal brains in the first and second trimesters also display significant changes in DNA methylation of the promoter regulatory regions of the genes involved in neurodevelopmental processes, such as *MEIS1* (homeobox gene), *NF1C* (nuclear factor 1 C-type), and *FAM49A* (carrying cytoplasmic fragile X-interacting superfamily sequences) [46]. Given that human fetal-specific epigenetic machinery and methylation modifications are identified in the proliferative zones of both the cortical and GE regions [45,55], inhibitory neurogenesis in the human fetal brain is likely to be influenced by epigenomic plasticity. However, little is known about how epigenetic mechanisms specifically regulate the GABAergic inhibitory neurogenesis for more protracted periods and what their differences are across species.

### 2.3. Vulnerability of Embryonic Neurogenesis to Environmental Risk Factors

Certain environmental insults (deficiencies in oxygen or nutrient levels, infections) are highly associated with neuropathological conditions [56,57]. Since oxygen levels are key in maintaining neurogenesis, hypoxia during the embryonic stage is associated with deficits in neurogenesis and significantly increases susceptibility to neuropathology, such as observed in white matter changes in hypoxic-ischemic encephalopathy [58,59]. For example, prematurity reduces the number of PV+ interneurons in the mouse cortex, which leads to a disruption of the inhibitory neural networks in the cortical upper layers [60]. Rabbits born at E29 (full-term, 32 days) show a decline in MGE neurogenesis [38]. Lastly, human neonates born before 32 GW exhibit fewer GABAergic interneurons in the white matter and subplate; the density of SST+ and calbindin+ (CB)+ interneurons are significantly decreased in the PFC in these premature cases [61,62]. Thus, defects in inhibitory neurogenesis can decrease the number of cortical interneurons, eventually disrupting the excitatory and inhibitory balance and leading to cognitive deficits, such as those observed in ASD [63,64].

Environmental risk factors during gestation (prematurity, maternal stress) can also affect the developing nervous system via epigenetic modifications. Pregnant rats with chronic stress, for example, show increased DNA methylation on the promoter of the glucocorticoid receptor (*GR*) gene in the hippocampus, leading to an alteration of social behaviors [65]. Interestingly, maternal depression during the third trimester of human pregnancy increases methylation on *GR* promotor regions in the newborn and is associated with altered cortisol stress responses [66]. These epigenetic mechanisms could serve as mediators for brains to adapt to environmental changes. The disruption of key epigenetic regulators, such as DNMTs and LSD1, has been implicated in neurodevelopmental disorders (NDDs). For instance, DNMT1 is abnormally overexpressed in GABAergic interneurons in the prefrontal cortex (PFC) of the patient with schizophrenia or bipolar disorder [67,68]. Patients with epilepsy, ASD, and intellectual disability (ID) have common mutations in the functional domain of DNMT3 [69]; KDM1 decreases the binding at the promoter of *SCN3A*, thereby misregulating the *PHF21A* gene in the brain of ID patients [70]. These data open the possibilities for the expanded use of enzymatic inhibitors, such as those that have been clinically used for cancer patients with abnormal expressed DNMTs or LSD1, for treatment in NDDs [71]. Indeed, recent neurological studies have provided evidence that epigenetic drugs may be implicated in aging, seizures, and neurodegenerative diseases, including Alzheimer’s disease (AD) and Parkinson’s disease [72]. For instance, aging brains exhibit an overall decrease in genomic DNA methylation, and AD brains show hypomethylation in the promoters of amyloid precursor protein (*APP*), presnilin1 (*PS1*), and beta-site APP cleaving enzyme 1 (*BACE1*) [73]. Considering that the epigenetic regulators are involved in the neurogenesis of GABAergic interneurons during early development, these inhibitors could potentially be an initial treatment for NDD targeting GABAergic interneurons. Furthermore, it can also expand clinical implications with specific targeted drugs that promote the proliferation of the remaining progenitor cells after the cortical damages caused by external insults, such as stroke or injury.

## 3. Prolonged Migration of Coritcal GABAergic Interneurons

### 3.1. Early Postnatal Migration of GABAergic Interneuron into the Forebrain

Unlike excitatory neurons, young GABAergic interneurons are generated in neurogenic niches removed from their cortical targets and must travel long distances to reach their final destinations [8,74,75] (Figure 1). Their migratory properties, including morphology, directionality, and interactions with extracellular structures, are extremely diverse depending on their cellular identity and guidance cues [76]. Recent data also show that, unlike excitatory neurons, interneurons continue to migrate into the cortex after birth. Cortical neuronal migration is present in the early postnatal mammalian brain and varies across species. Therefore, defining the nature and behavior of young migratory GABAergic interneurons that persist after birth is fundamental for understanding the final acquisition of their functionality.

Migratory streams of neurons are still observed in the mouse brain at postnatal day 15 (P15) [77]. Using the 5HT3a-GFP mouse line, Inta et al. showed that migratory neurons originated from the mouse CGE and targeted several regions, including dorsal cortical areas, the striatum, the nucleus accumbens, and the occipital lobe at P10. GFP+ cells expressed both doublecortin (DCX), a microtubule-associated protein found in migratory neurons, and CR, supporting them as immature, migratory, and inhibitory [78]. The gyrencephalic ferret brain harbors DCX+ GABAergic interneurons in robust streams at postnatal ages within the PFC, the dorsal posterior sigmoid gyrus (equivalent to ACC in humans), and the occipital lobe. At postnatal day 20 (P20), these DCX+ young interneurons co-expressed CGE-related markers, such as SP8 and SCGN [79], suggesting that the CGE specifically gave rise to this postnatal migratory stream.

Young migratory interneurons expressing DCX and polysialylated neural cell adhesion molecule (PSA-NCAM) have been reported in the human anterior forebrain. These cells are abundant at the LV walls and are organized into a dense structure that is connected to dispersed populations of DCX+ PSA-NCAM+ cells in the overlying white matter and developing cortex [80]. The DCX+ PSA-NCAM+ population closer to the LV expresses NKX2.1, LHX6, SP8, and COUP-TF2, supporting their origins within the human GE. These cells also express GABA, CR, CB, neuropeptide Y (NPY), and SST, supporting that the DCX+ PSA-NCAM+ population migrating in the human developing forebrain have an interneuron fate. This population is observed until several months after birth [80]. Another postnatal population in the human cortex defined by the expression of DCX and PSA-NCAM is the medial migratory stream (MMS), transiently observed during the 4–6 months after birth. The cells within the MMS and its target, the ventromedial PFC (vmPFC), express the interneuron subtype markers, such as CR and TH [81,82], suggesting that these neurons have similar origins and features to the olfactory bulb (OB) interneurons. Thus, GABAergic interneurons can continue to travel into the early postnatal cortex, though the duration can vary across species. Further research is required to understand the specific contribution of the GE subregions to this late migratory population and how their behaviors are regulated.

### 3.2. Diverse Migratory Behaviors of GABAergic Interneurons

Studies in the rodent brain suggest that interneurons in the developing brain require critical external cues for tangential and radial migration to their intended regions [83,84,85]. One of the key cues for interneuron migration is GABA signaling. During the tangential migration, cortical GABAergic interneurons exhibit different responsiveness to GABA depending on the functional expression of GABA receptor subtypes [86]. For instance, MGE-derived interneurons expressing the specific GABA A receptor show tangential migration into the neocortex, which is triggered by extracellular ambient GABA [87,88]. Cortical interneurons activated by the GABA A receptor in early postnatal mice show multidirectional motility within the tangential plane via changes in intracellular Ca2+ concentration [89]. Activation of the GABA A receptor by allosteric modulators, such as Zolpidem and Diazepam, facilitates membrane depolarization that activates voltage-gated Ca2+ channels in the tangentially migrating MGE-derived interneuron [87]. Considering that tangentially migrating cortical interneurons contract actomyosin and remodel microtubule through MLCK and CaMKK-AMPK pathways [90,91], it may be that pharmacological activation of the GABA A receptor boosts Ca2+ influx into migratory neurons to rearrange the cytoskeleton for tangential migration. Interestingly, the human parietal association cortex shows an increase in GABA A receptor-binding levels from mid-gestation into infancy [13]. It suggests that GABA signaling is dynamic and heterogeneous during the migratory process and could be a critical regulator for the migration of GABAergic interneurons. To transition from tangential into radial migration, interneurons need supportive structures as well as extracellular stimuli [92]. Interneurons traveling to the OB, for example, are surrounded by astrocytes aligned to form glial tubes along their path from the subventricular zone (SVZ) to the OB [93]. After arrival in the OB, individual interneurons detach from the RMS chains and are guided by blood vessels to integrate into olfactory circuits. However, the specific cues and structures needed in the postnatal migration of cortical interneurons, and how they might differ across species, remains unknown [94].

In the human infant brain, abundant DCX+ young neurons around the anterior body of the LV shape an eyebrow structure, which is termed as Arc [80]. This structure is organized by four distinct layers of DCX+ cells. Tier 1 is adjacent to the wall of the LV and is filled with tightly packed DCX+ cells. Tier 2 has looser densities of DCX+ cells with unipolar or bipolar morphology. Further away from the ventricular wall, the migratory morphology and directionality of DCX+ cells become more variable. Some DCX+ cells form clusters surrounding blood vessels, suggesting that vasculature could serve as a migratory scaffold here as well. The most removed region of the Arc, Tier 4 within the developing white matter, has huge clusters of DCX+ cells with triangular shapes and individually migrating DCX+ cells with long extensions directed to the overlying cortical plate [80]. In the MMS, DCX+ cells organize as chain-like structures and appear to migrate tangentially into the vmPFC [81]. Determining how these diverse structures are regulated in the forebrain and how they relate to the cognitive functions in the PFC or vmPFC is fundamental to understanding neuropsychiatric diseases with pathology in these specific regions.

### 3.3. Regional Cortical Vulnerability and GABAergic Interneuron Migration

The disruption of the migratory process by cytoskeletal defects or environmental insults can result in a decrease in the number of cells within focal cortical areas and result in abnormal neural connectivity. The abnormal migration of GABAergic interneurons has been associated with brain regional malformations seen in patients with neurodevelopmental disorders or periventricular nodular heterotopias [95,96]. Interestingly, the specific targets of postnatal cortical migration for interneurons appear highly correlated with the abnormal regions identified in patients with ASD or cerebral ischemia. Region-specific abnormalities in the PFC and ACC have been reported in brain imaging studies of patients with ASD, especially before the age of 2–4 years [97,98]. Altered neuronal numbers in the hippocampus and amygdala of children with ASD aged 2–3 years have also been observed [99,100,101,102]. Many studies have suggested that these regions may be associated with clinical symptoms in ASD [100,103]. For example, malformations of the PFC, ACC, and amygdala may lead to restricted social behaviors and defective communication [104,105,106]. Patients with periventricular leukomalacia (PVL) and white matter lesions (WML) from ischemic injury exhibit neuronal loss or microglial activation beneath cortical layer 6 and in white matter, areas that are highly populated by late migrating GABAergic interneurons [62,107]. These reports support that specific injury late in interneuron development can be linked to neuropsychiatric consequences and suggest regional-specific development of the structures involved in higher cognition [61].

## 4. Establishment of the GABAergic Interneuron Network

### 4.1. Circuit Maturation of GABAergic Interneurons for Refinement of Cortical Functions

Critical periods are a maturational stage of heightened plasticity, when neuronal connections are sensitive to experience and environmental influences, often affecting function irreversibly [108,109]. This time tightly corresponds to the development of thalamocortical connections as observed in the rodent primary sensory areas, including auditory, visual, and somatosensory cortices, or PFC [110,111,112,113]. Excitatory axonal innervations of the thalamus to the cortical layer 4 peaks during the postnatal critical period [20,110,114]. These neural connections induce long-term potentiation (LTP) and long-term depression (LTD) in the cortical pyramidal neurons, and they are thought to contribute to the modulation of sensory or cognitive information processing. Adequate sensory experience during the postnatal critical period is essential for normal neuronal connectivity, as shown by monocular deprivation experiments in mice where the covering of one eyelid throughout the critical period results in abnormal thalamic inputs into the primary visual cortex layer 4 and reduced cortical responsiveness to the covered eye [115,116,117]. Strikingly, these disrupted neural networks can maintain some aspects of their plastic traits for adapting to the external environment throughout life. For instance, patients who experience blindness from birth to adolescence and regain vision (via cataract removal) show more sensitive contrast ability than those of normal developing infants [118,119]. In addition, adults with amblyopia can demonstrate improved performance in perception on a visual task through repeated practice [119]. It has been suggested that the maturation of inhibitory neurons critically contributes to the formation of functional networks by driving the postnatal critical period. GABA synthetic enzyme (GAD65) knockout (KO) mice with low levels of GABA show no loss of dendrite spine pruning in layer 2/3 pyramidal neurons after monocular deprivation during the postnatal critical period (P25–26), which means GAD65 KO mice cannot open the critical period. However, increasing GABA concentrations in Zolpidem-treated GAD65 KO mice recovered the sensitivity to ocular dominance plasticity, like the normal mice [120,121]. In addition, the transplantation of inhibitory neurons into the mouse visual cortex was able to recreate a critical period, depending on the age of the cells transplanted and on the age of the recipient brain [122]. Therefore, it is of great significance to clarify the role of late-integrated GABAergic interneurons during the postnatal critical periods in regulating neural functional connectivity and plasticity.

The processes of inhibitory neurogenesis, migration, and network integration lag behind those of excitatory neurons, and their maturation periods of connectivity overlap with the critical period for postnatal plasticity [123] (Figure 1). This has been best studied in mouse primary sensory areas. The maturation of GABAergic interneurons, mediated by factors, such as neurotrophins (such as brain-derived neurotrophic factor (BDNF)), neuregulins, and orthodenticle homeobox 2 (Otx2), leads to a critical period, the basis for the balance between excitatory and inhibitory information for the sensory topographic mapping [124,125,126,127]. These maturation steps are also sensitive to the environmental changes encountered during the postnatal critical periods; altered hearing in the mouse significantly reduces the inhibitory synaptic components in the primary auditory cortex, and visual deprivation disrupts the interneural integration into excitatory pyramidal neurons [126,128]. Subsequently, the maturation of GABAergic interneurons increases the synaptic strength or activity of adjacent pyramidal neurons in the mouse primary visual and auditory cortex [111,129,130,131]. Thus, normal interneuron development is fundamental to refining the functional neural network and subsequent sensory processing [121,132,133,134].

Interneuron maturation is highly associated with the generation of beta-gamma oscillations, local brain activity in the range of beta (15–40 Hz) and gamma (30–80 Hz) frequencies, which are regulated by SST+ and PV+ interneurons, respectively [135,136]. This rhythmic fluctuation is observed in the primary visual cortex on visual tasks. In the macaque brain, the visual cortical areas show strong gamma oscillation during visual tasks, such as visual attention or perception [137], which is thought to be regulated by PV+ interneuron activation, as in the mouse visual cortex [136]. Similarly, the generation of neural networks mediated by GABAergic interneurons is also observed in the higher cognitive regions, such as the PFC. For instance, mice with disrupted development of MGE-derived interneurons display abnormal gamma oscillations in the PFC corresponding to impaired cognitive tasks. Optogenetic stimulation of the disrupted MGE interneurons in the PFC recovers 40–60 Hz oscillations and normalizes the impaired cognitive functions [138]. Thus, the normal development of MGE-derived interneurons critically influences the gamma rhythmical functional networks in the PFC and associated cognitive functions.

Recent studies in primates show a strong correlation between the timing of interneuron maturation in each brain region and the developmental order of higher cognitive areas. In the macaque brain, the timing of PV+ interneuron maturation happens sequentially, with cell density and axonal projections of PV+ interneurons peaking first in the primary visual cortex, followed by higher associative areas, such as the inferior temporal cortex and the PFC [129,139]. The human brain also has a developmental order for different cortical regions. Basic sensory pathways, such as smell, vision, and hearing are the first to develop rapidly, followed by language acquisitions and then higher cognitive function [140]. Given that both the human PFC as a major center for executive functions is one of the last regions of the brain to develop [141], and electroencephalography (EEG) recordings in the human PFC display an increase in gamma oscillations during PFC-related tasks [142,143,144], it strongly supports that extended maturation of GABAergic inhibitory circuits is associated with the acquisition of cognitive functions in larger, gyrencephalic animals, like humans.

### 4.2. Extracellular Matrix Regulation of the Inhibitory Circuit Plasticity

The perineuronal net (PNN) is an extracellular matrix (ECM) structure composed of proteoglycans and is preferentially expressed on mature PV+ interneurons in the mouse visual cortex, barrel cortex, PFC, and amygdala [111,145]. This structure, surrounding PV+ interneurons, appears at the end of the postnatal critical period within each region and reduces local plasticity. When PNNs are enzymatically digested in the mouse visual cortex, it promotes neuronal plasticity by re-establishing an early postnatal-like state. In addition, the removal of PNNs in PV+ interneurons in the mouse medial PFC leads to the disruption of the gamma rhythm [146]. Recent studies have reported that PSA-NCAM, which is typically viewed as an anti-adhesion molecule that interacts with the ECM and promotes neuronal migration, can also affect synaptic dynamics and regulate the plasticity of PV+ interneurons in the medial PFC under chronic stress [147] and structural plasticity in the visual cortex during the critical period after chronic monocular deprivation [148]. The deficiency of ECM components, such as Otx2, interrupts the opening of a critical period in the mouse primary visual cortex; the overexpression of Otx2 can accelerate the postnatal plasticity as well as PV+ interneuron development under visual stimulation [125]. The extracellular matrix around cortical interneurons thus may play a role in changing the features of plasticity in a region-specific fashion. Understanding the developmental timing and specific localization of PNNs will help determine how plasticity is regulated and potential avenues for therapeutic shaping of the cortical circuit.

### 4.3. Imbalanced Neural Networks Due to Interneuron Abnormality

An imbalance in cortical excitation (E) and inhibition (I), driven by interneuron dysfunction, is thought to underlie NDDs, such as ASD, schizophrenia, and epilepsy [149,150]. Since the E/I imbalance in the cortex is reflected in gamma-band oscillations, this altered rhythm is being investigated as a potential biomarker of patients with ASD [151]. Interestingly, patients with ASD or schizophrenia exhibited a lower frequency of gamma oscillations in the visual cortex, auditory cortex, motor cortex, and frontal cortex, which are areas regulated by the GABAergic interneurons for the acquisition of cognitive functions (see Section 4.1) [152]. Magnetic resonance spectroscopy (MRS) imaging in those regions from patients with ASD has shown significantly decreased concentrations of GABA neurotransmitters, and a postmortem study of brains from patients diagnosed with ASD shows a reduced number of GABAergic interneurons as well as the decreased genetic level of PV [153,154]. These studies connect the presence of functional networks with disrupted gamma rhythms in patients with abnormal cognitive functions and support the importance of a normal inhibitory network through development.

## 5. Maintaining a Population of Immature Neurons in the Adult Brain

After the completion of classical developmental stages, are there other mechanisms by which the neuronal circuits can maintain flexibility, even at older ages? Recent studies have supported a possible avenue via the presence of immature-like neurons [155,156,157]. These investigations revealed regional-specific populations of DCX+ PSA-NCAM+ cells in the adult gyrencephalic brain. These neurons were observed in cat, sheep, marmoset, guinea pig, and human brains, and lacked morphological features associated with migratory cells [158,159,160,161,162]. These cells are less prevalent in smaller, lissencephalic brains, thus associating them with evolutionary changes in larger brains. These cells are observed in cortical layer 2 of the frontal and temporal cortices but have also been identified in the amygdala of postnatal cat, sheep, squirrel monkey, and human brains [102,158,163] (Figure 1). Analyses in marmoset and human amygdala and frontal lobes showed that the proliferative marker, Ki67, was primarily expressed in oligodendrocyte precursors; few DCX+ cells in these areas expressed Ki67 [102,158]. This supports that DCX+ cells in the postnatal gyrencephalic cortex are in an immature, post-mitotic state. Immature cells have been reported in human frontal and temporal lobe samples at as late as 58 years of age [162]. In the young adult cat, rabbit, guinea pig, and sheep brains, these cells appear to have excitatory cell fates, given their expression of TBR1 [159,164]. DCX+ PSA-NCAM+ cells are found in the adult human and monkey cortices, but their cell identity is unknown [161]. In neonatal sheep, the amygdala contains DCX+ cells of mixed origins, with a 1:3 ratio of interneurons to excitatory neurons [158]. These studies suggest that immature cells are diverse in their cell fates and could vary regionally and across species. Further studies are needed to determine the origin and role of immature neurons, either excitatory or inhibitory, in regulating brain plasticity and function.

## 6. Conclusions

GABAergic interneurons have extended properties of production and maturation that are further amplified in larger, gyrencephalic brains. The postnatal incorporation of GABAergic interneurons in specific regions of the cortex may be a cellular mechanism for the formation of higher cognitive tasks in different species, such as in humans. Furthermore, the longer developmental period experienced by GABAergic interneuron may function as a mediator for adaptation to diverse environmental stimuli. Lastly, it provides potential clues for additional avenues for postnatal brain plasticity to maintain the brain in a more flexible stage for a protracted time after birth.

Understanding the protracted nature of GABAergic interneurons could uncover novel mechanisms and therapeutic opportunities for neuropsychiatric and neurodevelopmental disorders. These are conditions with limited treatments, stemming from our gap in knowledge regarding this important neuronal population, especially within the human brain. Determining the mechanisms that regulate in their long developmental timeline, especially in the early postnatal period, would allow for therapeutic intervention and modifications during infancy and beyond.

## Figures and Tables

**Figure 1 ijms-22-05113-f001:**
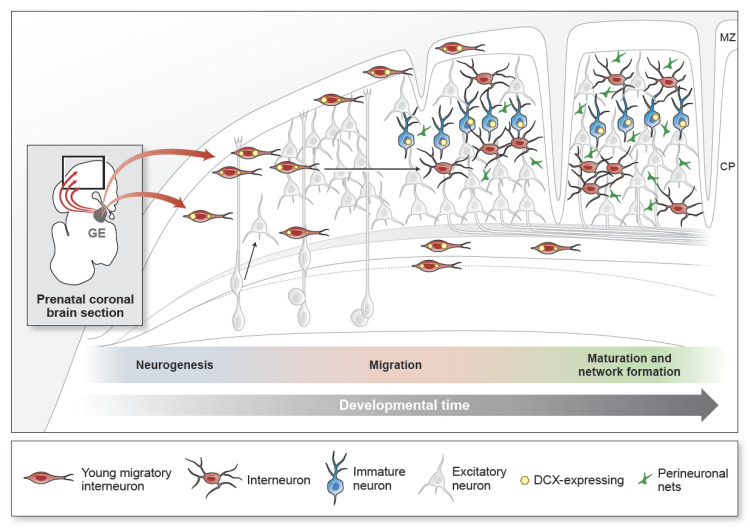
The developmental processes of cortical GABAergic inhibitory neurons. Inhibitory neurons (interneurons) contribute to 20–30% of the total cortical neurons. They are generated in the ventral embryonic brain in a region called the ganglionic eminence (GE) during mid- to late-gestation (schematic shows a representative coronal section of the prenatal brain). Excitatory cortical neurons, in contrast, are generated from radial glial progenitors at the end of the first trimester and establish the cortical plate in an inside–out manner (shown in gray). Interneurons migrate from the GE and appear in the cortex early in the second trimester. Young migratory interneurons (shown in red) have a leading process and express DCX (yellow) as they migrate into the developing cortical plate (CP). The CP is the precursor to the multi-layered cortical structure. Interneuron migration into the cortex extends to the early postnatal period in gyrencephalic brains. Interneurons in the cortex increase their morphological complexity and decrease DCX expression with maturity. In adulthood, the DCX-expressing population (shown in blue) remains in cortical layer 2. Their cellular identities and functions are unknown. Within the developing cortical circuit, synapse maintenance is influenced by sensory stimuli and mediated by extracellular matrix components, such as perineuronal nets (PNNs) (green). Synapses are stabilized as PNNs increase with age. Progression from prenatal to postnatal ages is represented by the grey arrow (developmental time) directed left to right. MZ, marginal zone; CP, cortical plate.

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
