# Peer review of "Implications of Extended Inhibitory Neuron Development"

_ijms, 2021, doi:10.3390/ijms22105113_

Round 1
Reviewer 1 Report
These authors have submitted the manuscript entitled ‘Extended development of inhibitory neuron for brain plasticity’ as a review article, and they explained interneuron development including GABAergic inhibitory neurons for the regulation of brain plasticity in this manuscript. This manuscript is interesting, but there are some critical points need to be answered for the publication to International Journal of Molecular Science.
- The order of the subtitles is not well understood. These authors wrote the development of GABAergic neurons in embryo (Section 2, Line 47-141), and postnatal period (Section 3, Line 142-236), but postnatal period have explained again in Section 4 and adult period is also adverted. And, Section 5 explained another topic (immature neurons) with different time and species. I might suggest that they need to reorganize the contents with detailed time dependent manner or any other reasonable methods.
- Explanation in Figure 1 looked very confusing. For example, ‘inhibitory neurons’ corresponding to ‘excitatory neurons’ are not shown in figure 1, and too many neurons are not systemically classified (migratory interneuron vs. interneuron? Immature neuron vs. excitatory neuron?). In addition, whether the direction of progress from left to right in the picture means the passage of time or the position of the brain, the clear intention cannot be read.
Reviewer 2 Report
The manuscript presents interesting information concerning neurogenesis of GABA-ergic neurons, their proliferation and migration. However, the title of the manuscript suggests that the reader will get information concerning neuronal plasticity, whereas all we have is, undoubtedly interesting, information about formation of GABA-ergic network. There is no information how formation of GABA-ergic network affects neuronal plasticity which mainly consists in modulation of synaptic connections. Therefore I would suggest changing the title of the manuscript and making appropriate changes to the text.
I also suggest the authors a careful reading the text as it is full of type mistakes that change the meaning of the sentences (example - perineuronal net not “perineural net”).
Reviewer 3 Report
Kim and Paredes present a nice review about the "Extended development of inhibitory neuron for brain plasticity.". The review is rather complete, well written and clear. However there are some points that can be improved/modified to make it even more complete if the authors agree.
1 - The title can lead to confusions. It could include some mention to GABAergic neuron that seem to be the main topic in the text.
2 - The neurogenesis regulation by KDMs, LSD1 or DNMTs could be better explained. A more detailed explanation would be useful for the potential readers to understand this key mechanism. Now it is quite difficult to follow.
3 - The authors stated that "defects in inhibitory neurogenesis can decrease the number of cortical interneurons, eventually disrupting the excitatory and inhibitory balance and lead to the deficits observed in neurodevelopmental disorders (NDD) such as Autism Spectrum Disorder (ASD) and epilepsy" Interestingly there have been attempts in the last years to found inhibitors of LSD1, DNMTs or lysine demethilases aimed to treat aging related diseases as well as other neural disorders like epilepsy, seizures, etc. It would be interesting if the authors can discuss about it. Can these strategies help to repair the cortical network after birth (as it is exposed in the conclusions that the processes of proliferation and migration of GABAergic interneurons in gyrated brains can help to that).
4 - How GABAr agonists/antagonists (direct or indirect via inhibition of other proteins of the pathway) affect neuronal migration?
Reviewer 4 Report
This is interesting review article to be accepted after a minor revision.
The Authors should provide a more detailed introduction to explain in detail the role and importance of GABA-ergic neurons in the central nervous system. This should explain to the Readers, why it is of importance to study mechnisms of cortical plasticity.
Round 2
Reviewer 1 Report
These authors revised the manuscript extensively, and, compared to the previous version, this version helped to improve the explanation and understanding of the contents. However, they need to refine figure 1 extensively to explain the migration process more clearly. For example, legend in figure 1 showed repetitive contents and typos (...Young migratory interneurons (bipolar, shown in red) expressing DCX (yellow) migrate tangentially radially along a radial glia scaffold radially to integrate into the cortical circuits. The young migratory interneurons in gyrencephalic brains continue to migrate into the cortex during the infancy (See 2.??).." The layer of cortex might also be marked in the figure, and I did not see any difference between "Migration" and "Maturation" phases.
Author Response
We thank the reviewer for their critical reading and thoughtful comments.
As the reviewers suggested, we have modified the legend in figure 1. When young interneurons migrate from GE to the cortical plate (CP), they have a leading process and express DCX (Migration phase). Once arrived, young interneurons are differentiated into mature interneurons having a complex morphology and marginal expression of DCX (Maturation phase). To clarify this assertion, we have changed the legend.

Reviewer 2 Report
The revision of the manuscript is satisfying and in my opinion the manuscript is suitable for publication.
Author Response
We appreciate for reviewer's efforts in the review process, and we are grateful the reviewers for providing detailed and constructive comments.
Round 3
Reviewer 1 Report
These authors made their efforts, and now this manuscript is ready to be published to International Journal of Molecular Science.